# Are we ready for self-sampling for cervical cancer screening? Insights from service providers and policy makers in Nepal

Saki Thapa[1]*, Amit Timilsina[2], Bikram Bucha[1,3], Swastika Shrestha[1], Safal Kunwar[1], Raghu Dhital[1], Gillian Holdsworth[4]

1 Research Department, Birat Nepal Medical Trust (BNMT), Kathmandu, Nepal, 2 Research and Community Development Center, Kathmandu, Nepal, 3 Department of Global Public Health, Karolinska Institutet, Stockholm, Sweden, 4 Britain Nepal Medical Trust, United Kingdom of Great Britain and Northern Ireland (UK), London, United Kingdom

* saki.thapa@gmail.com

**Data Availability Statement:** All the data required for this study has been included in the manuscript. The manuscript contains comprehensive descriptions of the qualitative data, including

## Abstract

Cervical cancer is the leading cancer among women in Nepal, but the country has very low screening rate, with only 8.2% of women being screened. In recent years, a self-sampling kit for testing for the human papillomavirus (HPV) has been developed to allow self-sampling and enable early detection of cervical abnormalities. This kit enables women to collect cervical samples without the need for trained healthcare providers or healthcare facilities. Nevertheless, there has been a notable absence of scientific studies to analyse the feasibility and acceptability of self-sampling for cervical cancer screening in Nepal, particularly from the perspective of various service providers. The qualitative research method used semi-structured in-depth interviews and key informant interviews with healthcare providers, online service providers, and policymakers. These interviews were conducted in person with 20 participants until data saturation was achieved. Thematic analysis was performed where the translated data was coded inductively using NVivo 12. The majority of the participants identified the self-sampling method as an alternative sampling option for detection of cervical abnormalities/cancer in Nepal. Barriers to self-sampling included a low level of knowledge and information, the cost of the self-sampling kit, unclear information regarding self-sampling process and concerns about inaccurate results among women and girls, who are the end users. Similarly, factors such as knowledge and information regarding the self-sampling technique, accessibility of available services and information, and national self-care guidelines and policies for cervical cancer screening were identified as facilitators for self-sampling. It is crucial to have evidence-based discussions, especially regarding the effectiveness of a self-care approach in cervical cancer screening and help create a supportive policy environment for implementing self-care strategies in Nepal. Furthermore, disseminating education and information nationwide through campaigns to raise awareness about self-sampling is essential among beneficiaries for the scaling up of self-sampling for cervical cancer in Nepal.

themes, codes, and illustrative quotes, that directly contribute to the main results or conclusions of the study. Additionally, it includes context and details relevant to the qualitative data and analysis.

**Funding:** This research work by Birat Nepal Medical Trust, Nepal ( BNMT Nepal) under the project Horizon was supported by the Britain Nepal Medical Trust, UK ( BNMT UK). The funder has no role in study design, data collection and analysis, decision to publish, or preparation of the manuscript but GH was involved in review of the manuscript once it was prepared.

**Competing interests:** The authors have declared that no competing interests exist.

## Introduction

Cervical cancer is ranked as the top ten most common cause of cancer incidence and mortality in women worldwide, with more than 90% of the new cases occurring in lower middle income countries such as Nepal in 2020 [1, 2]. The situation is no different in Nepal, where cervical cancer ranks as the second most common cancer among women and third most common cause of cancer death in women aged 15–44 years [2]. It is estimated that around 2.0% of women in Nepal are currently infected with cervical Human Papillomavirus (HPV-16/18) [3]. If left untreated, a persistent HPV infection of the cervix causes 95% of cervical cancers [4]. When diagnosed early and managed effectively, cervical cancer is one of the most successfully treatable forms of cancer. The World Health Organization (WHO) recommends women should be screened for cervical cancer every 5–10 years, starting at age 30 while screening should be undertaken every 3 years for women living with HIV, starting at age 25 [4]. The global strategy recommends that women undergo at least two high-performance HPV tests in a lifetime: once by age 35 and another by age 45, reaching at least 70% women screened by 2030 [4, 5].

The WHO also recommends a well-organized system for effective cervical cancer screening, diagnosis, treatment, and follow-up [6]. The standard diagnostic sequence for cervical intraepithelial neoplasia (CIN), a premalignant lesion, involves cytology (Pap smear), colposcopy, biopsy, and histological confirmation [6]. However, implementing a national screening programme requires a strategy, equipment and skilled human resources [6]. The Nepal national guidelines for cervical cancer screening and prevention (CCSP)recommends screening with the visual inspection of the cervix with acetic acid (VIA) test [7]. The initial target was to screen at least 50% of women in the age group of 30–60 years, by 2015, revised in 2017 to 70% [5, 7]. However, only 8.2% of women aged 30–49 years were screened for cervical cancer in Nepal by 2019 [8] as Nepal lacks a structured national screening program for cervical cancer [9]. In Nepal, cervical cancer screening is offered through two main approaches: population-based outreach health camps in communities, which are organized by hospitals in collaboration with government and non-government organizations, and outpatient clinical services, which are mainly provided by tertiary care centers [10]. VIA test that the Government of Nepal has implemented for cervical cancer prevention following WHO recommendations [6, 7] are available for free at government health institutions in 64 out of 77 districts for women aged 30–60 years. While people pay fees for Pap smear in locations where the necessary technical and laboratory facilities exist, a systematic review and meta-analysis suggests low cervical cancer screening uptake with 17% of the cervical cancer screening occurring in hospitals and 16% of the screening take place at the community [9].

In recent years, the development of kit for HPV self-sampling allow patients to easily collect samples themselves without the need for trained personnel or infrastructure for pelvic examinations [11]. The global experiences indicate that self-collected samples, when mailed and tested for HPV is an effective method to increasing cervical cancer screening rates among women [12]. This approach has the potential to reduce cervical cancer incidence and mortality in women at risk of developing cervical cancer [13].

In the context of Nepal, the Government of Nepal emphasizes the importance of self-sampling in reaching larger populations in the reference manual for cervical cancer screening in Nepal 2015 but highlights potential difficulty of follow up with treatment for test-positive clients [14]. Despite the existence of feasibility studies for HPV self-sampling methods, the promotion of self-sampling has not been extensively incorporated into the policies or programs or implemented as a pilot or national program of Nepal [15]. Health care providers, online service providers and policy makers are key to facilitate the discussion and the perspective of health care providers, online service providers and policy makers regarding the feasibility and

implementation of self-sampling is crucial which has not been explored and researched in Nepal. This study aims to explore and understand the perceived feasibility and acceptability of the self-sampling method, along with its barriers and facilitators among health care providers, online service providers and policy makers in Nepal.

## Methods and materials

### Study design

The exploratory study design using qualitative research methodology [16] was employed to understand the barriers and facilitators for introducing the self-sampling method for cervical cancer screening in Kathmandu, Bhaktapur and Lalitpur districts of Nepal.

### Study setting

The study site for this study was the urban area of Kathmandu valley (Kathmandu district, Bhaktapur district and Lalitpur districts). Due to the presence of tertiary level health facilities, digital/online service providers, and federal-level policymakers who are based in Kathmandu valley, it was purposively selected for this study. In Nepal, most tertiary hospitals provide cervical cancer screening services [10].

### Participants and sampling

Purposive sampling was implemented, and participants with knowledge and experience working in cervical cancer screening and self-sampling (health care providers, online service providers and policy makers) were included in the study to meet the objective of the study. The purposive sampling is extensively used sampling method in qualitative study which involves identification and selection of information rich and experienced participants to provide information to help answer the research question and objective [17].

The study population for this research were individuals 18 years and above, health care providers and policy makers with experience and expertise in cervical cancer screening, self-sampling of cervical cancer screening, treatment, and/or policymaking (Table 1). In-depth

**Table 1. Characteristic of the participants.**

| Characteristic | Number (N = 20) n |
|---|---|
| **Gender** | |
| *Male* | 7 |
| *Female* | 13 |
| **Occupation** | |
| *Gynaecologist* | 4 |
| *Oncology Nurses* | 4 |
| *Online service providers* | 4 |
| *Pharmacist* | 4 |
| *Government officials from Department of Health Services (DOHS)* | 4 |
| **Affiliation** | |
| *Tertiary general hospital* | 1 |
| *Tertiary speciality hospital* | 1 |
| *Cancer Speciality hospital* | 2 |
| *Pharmacies of tertiary general hospital* | 4 |
| *Online services with medical supplies* | 4 |

interviews were conducted among healthcare providers (gynaecologists, oncology nurses) and online service providers (online delivery services, pharmacists). Additionally, Key Informant Interviews were carried out with government officials from the Department of Health Service, Ministry of Health and Population, aiming to gain insights into their perceptions regarding the feasibility and acceptability of self-sampling for cervical cancer in Nepal.

## Ethical consideration

The research team obtained ethical approval from the Nepal Health Research Council (Registration no: 299/2022) on 9th August 2022 before conducting the research. The preliminary participant list was developed by Birat Nepal Medical Trust together with the Department of Health Services. Participants received an information sheet outlining the study's goals and ethical considerations. All participants willingly agreed to take part in the study and consented in written form to participate in the study.

## Data collection tools and procedures

A semi-structured interview guideline was developed for the in-depth interviews (IDIs) and key information interviews (KIIs). The interview guidelines were pretested among 4 participants for face validation and the pre-tested data has not been included in the final analysis of this study. Between September to November 2022, a total of 20 face-to-face interviews were conducted, with no repeat interviews. We conducted 16 IDIs with health care providers and online service providers, and 4 KIIs with government representatives at their workplace. For IDIs, we interviewed four gynaecologists, four pharmacists, four oncology nurses, and four online delivery service providers. Similarly, four representatives from the Department of Health Services (DOHS) participated in KIIs. The IDIs and KIIs were conducted in the Nepali language and were recorded in the recorder. Each KII note and voice recording were immediately transferred to cloud storage to avoid data loss. The average interview time was 53 minutes.

The trained field researchers, along with principal investigator ST, conducted IDIs and KIIs with the study participants. The field researchers hold bachelor's degree in public health while ST holds master's degree in public health. The field researchers have received training in qualitative research and research ethics, and have sound experience conducting qualitative interviews. The field researchers were also oriented by ST regarding the objectives of the study, data collection tools and techniques, data management and ethics. Female researchers ST and SS are affiliated with Birat Nepal Medical Trust (BNMT) and both had previous experience working on a cervical cancer screening study.

The researchers established a good rapport with the participants before conducting the interviews to make them comfortable to share their opinions. To ensure the researcher understood what participants were trying to explain, the researchers summarised the key points with the participants after each interview. Reflection on each interview was undertaken before proceeding to the next interview with different participants. No transcripts were returned to the participants for comments or correction. A rich and deep account of the participants' experiences was achieved through probing and through observation notes taken by the researcher. Data collection continued until reaching the point of saturation, where no new information [18]. The multiple data collection method from various sources has been used to triangulate the findings of the study, memo of the data collection and de-briefing sessions has been conducted to minimize the respondent bias and social desirability bias [19].

### Data processing and analysis

The team followed the Consolidated Criteria for Reporting Qualitative Studies (COREQ) guideline for conducting and reporting the findings of this study. The interviews were initially transcribed word-to word in the Nepali language and later translated into English language. The transcript were read and re-read by all the researchers and the translated documents were coded using Nvivo 12 software, qualitative data management software. Initially 62 codes were developed for each transcript using inductive coding and a codebook was developed. The researchers of this study discussed the codebook together, grouped the codes to develop sub-themes and themes presented. The detailed description is provided in Table 2. Thematic analysis was carried out to interpret the data and to present the findings of this study in the form of themes and sub-themes have been presented describing the subthemes and themes supported by relevant representative excerpts. ST and AT verified that data presented aligned consistently with the study findings.

## Results

A total of three themes and seven sub- themes emerged from the codes, as illustrated Fig 1.

The themes and respective subthemes include: Perception regarding self-sampling; Feasibility and acceptability barriers to self-sampling (Low level of knowledge and information, financing self- sampling kit for the women, unclear sampling process, accuracy, and quality of sampling) and feasibility and acceptability facilitators to self-sampling (knowledge and information regarding self-sampling technique, accessibility and availability of services, national self-care guidelines and framework for cervical cancer screening).

**Table 2. Code tree to showcase codes, sub-themes and themes.**

| S. N | Theme | Subtheme | Code |
|---|---|---|---|
| 1. | Perception regarding self-sampling | N/A | Hesitation to use service, Geographical preference, Literacy level among community members, Effectiveness of services, Acceptability among participants, Accuracy of testing, Use of new technology, Convenience to use service |
| 2. | Feasibility and acceptability barriers to self-sampling | Low level of knowledge and information | Shyness among participants, Perceived level of pain among participants, Perceived breach of Privacy, Lack of confidentiality among client, Comfort level with services, Stigma related to cervical cancer screening, Awareness among client, Prior experience of visiting health services, Education level of participants |
| | | Financing of self-sampling kit for the women | Cost of services, Insurance coverage, Affordability of services, Out of pocket expenditure, Financial hardship, |
| | | Unclear sampling process, concern of accuracy and quality of sampling | Unclear screening process, Unclear process to collect sample, Transportation of sample, Unclear process at Lab, Distribution through Pharmacy, Unclear Storage process, Distrust towards online service provider, No information regarding collection centres, Delivery of report, Perceived accurate result |
| 3. | Feasibility and acceptability facilitators for self-sampling | Knowledge and information regarding self-sampling technique | Information in local language, Demonstration, Orientation to client, Campaigns conducted, Confusion among clients, Confusion among online service providers, Awareness among policymakers, Need for education and curriculum, Information through social media |
| | | Accessibility and availability of services | Online Service, Service accessed through pharmacy, Service availability in hard-to-reach area, Availability of health care provider, Training for service providers, Advocacy for accessibility of kit, Provision of clear instructions |
| | | National self-care guideline and framework for cervical cancer | Accountability of policy makers, Commitment from policy makers, Guidelines regarding cervical cancer self-sampling, Policy regarding cervical cancer self-sampling, Provision of Human Resources, Training provision for health care providers, Role of government in planning, Implementation of program, Verification of self-sampling process, certification process, Coordination for program implementation, Research and evidence generation, National programme for self-care of cervical cancer |

**Perception regarding self-sampling**

- Feasibility and acceptability Barriers to self-sampling
  - Low level of knowledge and information
  - Cost of self- sampling kit for the women
  - Unclear process between sample collection and delivery of the results
  - Concern of accuracy and quality of sampling and results

- Feasibility and acceptability facilitators for self-sampling
  - Knowledge and information regarding self-sampling technique
  - Accessibility and availability of services
  - Inclusive policy and national strategy

**Fig 1. The description of themes and sub-themes.**

## Perception regarding self-sampling

Health care providers and policy makers reported that self-sampling for cervical cancer screening is feasible and acceptable to women in Nepal. Self- sampling is seen as acceptable in terms of privacy and confidentiality and easy to use for women.

> "Self-sampling must be comfortable for women, and it also maintain their privacy and confidentiality. Women who are educated and have knowledge about it, then they can easily use this same as they use a pregnancy kit."(Oncology nurse-3)

> "There may be more educated people living in Kathmandu valley. It would be easier to make them understand and teach them. Once they understand it properly, they would surely go for a self-sampling method. If we only take Kathmandu valley, considering its education rate and given the fact that the people in Kathmandu come for regular screening also, self-sampling sounds easier for them. They do not have to go to the hospital so they might accept this method in my opinion."(Representative DOHS -1)

Health care providers highlighted that women often feel embarrassed and shy if they require a pelvic examination and avoid visits to doctors due to concerns about their personal privacy, painful examination and potential harm to their reproductive organs and that provision of self-sampling approach could help address the issues of personal privacy and shyness.

> "They feel shy and uncomfortable in the beginning. Some think that it will be painful and they won't be able to work for days after this check-up. Since we have to counsel them when we say that we are about to take out swabs from the cervix area, they feel scared that their uterus will

*be hurt. But we give counselling and assure them that the procedure will be easy, and then they feel comfortable". (Gynaecologist 1)*

Most of the study participants of this study identify self-sampling as an acceptable and potential alternative approach for cervical cancer screening in Nepal. The study's participants believe that self-sampling is perceived to be easier to implement in urban areas due to internet penetration, high literacy levels, convenience, assurance of privacy and acceptability among young girls and women but are sceptical about its implementation in rural areas.

*"The people in urban cities are mostly aware and educated. Not only in Kathmandu valley but also in other urban cities like- Narayanghat, Butwal, and Pokhara. If we promote it and inform about this new method then I believe people would perform self-sampling easily". (Gynaecologist- 4)*

Spreading positive messaging and advocacy about self-sampling will contribute to an increased endorsement for self-sampling and increased importance of self-sampling among women and girls as suggested by one policy maker:

*"If the number of tests for cervical cancer self-sampling increases and if 1–2 people do it, the message would flow and other people would also do it". (Representative DOHS -4)*

In terms of feasibility, online service providers and policy makers reported that self-sampling is feasible in Nepal. The self-sampling is seen feasible mostly in terms of delivery of services, and level of education mostly in Kathmandu valley. One of the online service provider also reported that if the self- sampling product is available in the market then, it is feasible to deliver online in Nepal.

*"It is feasible to implement in Kathmandu valley because it includes the capital and the most educated city. It can be implemented quickly if government of Nepal and health related organizations works in coordination". (Pharmacist- 1)*

*"If we start the self-sampling system and if women could perform the sampling as advice, she can buy the kit, take the sample and submit it to the lab. This will likely increase the cervical cancer screening rate in Nepal" (DOHS-4).*

*"Self- Sampling kit is not a very big package, as it is a small package to deliver. We have a very large group of women who buy online other women products with us. If we have the self- sampling kit we can easily deliver it at home like any other product."(Online service provider-1)*

The majority of the health care providers and policy makers in this study express that self-sampling offers comfort, convenience, privacy and is perceived as a significant benefit for young girls and women. However, some of the participants of this study suggest that self-sampling is not feasible and acceptable in the context of Nepal, preferring screening by health care providers, and anticipating potentially low participation from the service users.

*"It is an advantage for women who feel shame going to hospital. It is beneficial for those who are shy and hesitate to do it annually. It saves time to visit doctors in the hospital and also their privacy is maintained."(Online service provider- 2)*

*"I think a certain proportion of people will prioritise self-sampling but certain people will reject it as well". (Representative DOHS- 2)*

One health care provider expressed a negative view regarding the need for self-sampling, suggesting that women can visit health facilities for HPV DNA testing.

*"It is hard to do that. How to focus on using this thing (self- sampling kit), for example, if you want to use it within Kathmandu valley, even if you go to a hospital, co-testing HPV/DNA test is available everywhere. So why does a person buy online and use it? It does not seem to be very effective". (Gynaecologist 3)*

## Feasibility and acceptability barriers to self-sampling

**Low level of knowledge and information.** Health care providers and policy makers report that the low level of cervical cancer screening is because of lack of knowledge and awareness among women. Cultural factors, shyness and hesitation, are highlighted as barriers by health care providers, contributing to the low level of cervical cancer screening among women in Nepal in the health care setting. Health care providers noted that the current screening services for cervical cancer are not acceptable to women because of inconveniences in health facilities. Gynaecologists and nurses report that women typically visit hospital or healthcare providers only after they experience symptoms.

*"The first thing is they hesitate to come to the hospital and those who come, they come only if they have other problems such as skin problems or ear problems. But when asked to show that private part, they are very shy"- (Gynaecologist- 3)*

*"There is an issue of privacy with people in the hospital. People feel uncomfortable exposing their private parts in front of the doctors. So I think this makes a difference."- (Pharmacist- 4)*

Since, the knowledge is already low regarding general cervical cancer screening, health care providers and policy makers reported that the low level of knowledge and awareness regarding self- sampling method could be barriers for acceptability and utilisation of the services.

*"First of all, people are not educated about the importance of screening. When people do not know that screening is important, why would they think about self-sampling? If I know that screening is necessary for my health, only then there would be a question to choose whether it will be done by doctors or by myself." (Representative DOHS- 1)*

Throughout all IDIs, service providers (health care providers and online service providers) emphasize the importance of raising awareness amongst women about cervical cancer screening and the available methods within the country. Additionally, some health care providers were unaware of the option of the self-sampling method for cervical screening.

*" If you give them this thing (self- sampling kit), they will not know how to open this or use this. Even I did not know how to open it before you showed it to me. You have to go campaigning from place to place. Women should know what this thing is and how to do it because most of the females in our Nepal are not educated." (Gynaecologist 1)*

**Cost of self- sampling kit for the women.** Heath Care providers believe that if the cost of a self-sampling kit is high, young girls and women might be unwilling to purchase and use it for screening. Some health care providers also emphasize the need to strengthen service

delivery in addition to the provision of a low cost kit. They stress the importance of making it cost effective to ensure acceptability among women in Nepal.

*"The price of self- sampling cervical cancer screening kit should be reasonable. I remember the urine pregnancy test kit used to cost 150–200 Rupees before but now it is available for around 35 rupees. So it might be costly in the beginning. But more than cost, awareness is the important factor here. If I am aware that I should do screening, I will arrange some money anyhow. There is hardly such a population who cannot spend 400–500 in the urban setting". (Gynaecologist 4)*

**Unclear process between sample collection and delivery of the results.** An unclear screening and sampling process was identified as another potential barrier by the health care providers and online service providers, particularly the process of sample collection, transportation to laboratory and delivery of results.

*"Self- sampling is possible, but after collecting the sample, we need to know how much time it should take to deliver it to the lab. Whether we have to collect the sample as soon as we deliver the kit to them or women take their own time to collect it and call us and then our rider will go and bring it to the lab. We need to know this. If the lifetime of the sample is up to 24 hours, then we can take it to the lab". (Online service provider- 1)*

*"So women should also know how to use it. They should be able to maintain sterility that is important too. They should know about the number of hours within which the sample should be taken to the lab, know about the temperature they have to maintain while transporting the sample from home to the lab. All of this should be managed and people should be aware. – (Gynacologist-4)*

**Concern of accuracy and quality of sampling and results.** A representative from Department of Health Services (DOHS), along with one gynaecologist and one oncology nurses, expressed doubt about the self-sampling approach through online services or through pharmacy. They perceived that self-sampling might be challenging for women particularly the need to follow the instructions, potentially leading to a dilemma about correct ways of self-sampling and the risk of performing the procedure incorrectly. They identify a lack of awareness or knowledge or literacy regarding self-sampling approach as potential contributing factors.

*"I think the women will be confused whether they have to take the samples from outer or inner parts, what is the best time to take the samples so that we can get quality samples? We will have to bring them in groups and do some teaching". (Oncology Nurse-3)*

The majority of the participants of this study were sceptical about the transport of sample after collection (self-sampling) to lab unless there is established mechanism for online delivery of lab samples for analysis.

*"In the beginning, if there is no knowledge, there is a chance of incorrect sample taking. Also women may not know how and where to send the self-collected sample. The testing labs may be far away and the report may not come on time. The important thing is how affordable it is or not. If it is not affordable, there may be less use and if there is no awareness, quality samples may not come". (Oncology Nurse -3)*

### Feasibility facilitators for self-sampling

**Improve knowledge and information regarding self-sampling.**   Health care providers and online service providers also reported that acceptability of the self-sampling by women depends on the knowledge of the sampling technique, therefore, it is needed to increase the awareness and knowledge of self-sampling among service users.

*"The main thing for promotion is to spread public awareness of this cervical cancer test kit among people about how to do screening at home, who can test it, where to send the sample". (Online service provider-1)*

Community-level awareness-raising activities are important to share information about the importance and process of self-sampling among women and to address perceived stigma, discrimination, hesitation and shyness regarding uptake of the self-sampling method.

*"Now that this (self-sampling) has been an alternative method, first we have to explain what it is (to community). You have to campaign from place to place. Women should know what this thing (self-sampling (is and how to use it because most of the women in Nepal are not educated (about self-sampling process) and they feel shy and hesitant". (Gynaecologist 3)*

Media, including social media, can be an important tool for marketing and creating awareness about cervical cancer screening with self-sampling kit among women in both rural and urban communities. Since most urban communities have access to social media and the internet, information available through these media are more appropriate for raising awareness as well as for improved acceptance of self-sampling approach in the urban setting.

*"We have a website as well as a mobile app. Those who are technology friendly can order from the website or mobile app such as Whats's App, Viber or even through email. But those who can't order from websites or mobile apps can order by a phone call.". (Online service provider- 4)*

**Accessibility and availability of services.**   Health care providers in the study emphasize the need to prioritize, implement, expand and extensively promote self-sampling for cervical cancer screening. They believe it could be effective in increasing self-sampling, based on the previous experience such as availability to pregnancy self-test kit.

*"If the self-sampling is easily accessible as the pregnancy self-test kit, everyone would do it. I think people are not concerned about money these days even if it is expensive. I am hoping it will be affordable if you make it available through pharmacies, I think people would do it". (Gynaecologist- 4)*

The health care providers in the study highlight the importance of promotion and endorsement of the self-sampling approach at hospitals and health posts. Alternatively, health care providers also suggest important role of pharmacies in promoting the self-sampling approach among its clients, as suggested by Nurse and Pharmacist:

*"One of the mediums is the hospitals, if it is available in hospitals, they (women) can get it from there. The other is the local pharmacies. I am not sure if it is available online right now. If yes, then it can be accessed online as well". (Oncology Nurse -4)*

*"If it is made available in pharmacies, the pharmacy staff there can provide more information and boost their confidence too, that is ' why it seems easier to deliver through the pharmacy itself". (Pharmacist- 4)*

Online service providers share that service users are familiar with online health care due to the shift in health seeking behaviour from physical consultations to online consultations during COVID-19. Online Service providers suggest that online delivery is feasible in Nepal, especially in urban setting, due to internet facilities and availability of courier delivery services. However, online service providers are uncertain if the service user in more rural areas will be interested in the services due to low internet coverage and added delivery cost for the self-sampling kit.

*"Online purchase of self-sampling kit is possible in urban areas. If the client pays the transportation charge, we will provide it. We don't know the feasibility in the villages. But in the city, women do not shy away to buy the medical product online from our experience". (Online service provider- 2).*

A policymaker also supports inclusion and mobilisation of the private sectors in promoting and providing self-sampling cervical cancer screening services.

*"If there are clients who are willing to use the paid for services, it could be done by engaging with the private sectors. We only talk about the public sector but private service providers or online service delivery can provide services by charging the fee". (Representative DOHS- 2)*

Even though online service provider are ready to provide and deliver self-sampling kit, they also emphasize the need for trained online service providers and health care providers to counsel and support client for accurate use of self-sampling kit. A pharmacist also mentioned that providing information in the kit on how to use it would be helpful, but also signposting women to visit health professionals for assistance in understanding the kit. Developing the information leaflet or instruction manual for self-sampling in Nepali language will help women understand and follow the instructions.

*"It is feasible, but even if it is done online, it seems that it would be better to involve only trained personnel. Even those who have studied Bachelors in Business Administration (BBA) involved in marketing line say they are in medicine, but they do not have knowledge about it, so they cannot give certain information. That is why it is better if people in the medical field with medical knowledge offer the service online". (Pharmacist- 2)*

*"Adding information in the kit will make it easier. No matter whether you give awareness through videos or not, the manuals must be there. If it is in Nepali rather than English, that is even better. People can read and that will make it easier. Even I would find it easier to read if it was in Nepali. So these instructions are a must." (Pharmacist- 4)*

**Inclusive policy and national strategies.**    Policy makers emphasise the need for guidelines and a framework for self- sampling of cervical cancer screening or inclusion of self-sampling of cervical cancer screening in the existing national guidelines of Nepal. The policymaker highlights that discussion for inclusion in the national guideline has been initiated but the endorsement has not been completed yet. Participants also share that the guideline will help both

public and private sectors to engage in effective service delivery and improve the acceptance of cervical cancer screen in Nepal.

*"The family planning section (of MOHP) is advocating for a national screening program and the processes to establish it are already in place. They are yet to decide on things like how to carry out screening, which health workers to be mobilised and trained, dispatch, collection and lab testing sites etc. I believe the paperwork has already been completed but there is a dilemma as to whether to adopt the self-sampling approach or not. More research like this is crucial to provide us with evidence regarding adoption and acceptance of self-sampling for strategic and policy decisions".* (Representative DOHS-3)

Current national policy and guidelines do not envision the inclusion of the private sector in cervical cancer self-sampling. One policymaker highlighted the need to include the private sector in policy provision to offer the self-sampling service. They also suggested implementing a quality control mechanism to avoid variation in usage of kits and process for self-sampling, applicable to both public and private service institutions and providers.

*"In respect of quality control, we have to create a control mechanism in parallel for both private and public sector. For this we need guidelines to include and engage the private sectors".* (Representative DOHS- 2)

## Discussion

This study explores the feasibility and its acceptability of HPV self- sampling in Nepal among health care providers, online service providers and policy makers. The findings of this study present the perspectives of health care providers, online service providers and policy makers on the feasibility and acceptability of self- sampling for cervical cancer screening in Nepal. The findings of this study highlights convenience, and confidentiality, as key characteristics which help in the adoption of self-sampling among the women and girls. These factors might contribute to an increase in cervical cancer screening uptake in Nepal and align with similar benefits reported in previous studies [20–23].

In alignment with the standard of care, there is evidence that establishes the importance of self-sampling of HPV, which can increase cervical cancer screening uptake [12]. Although health care providers, online service providers and policymakers find self-sampling to be a feasible process for increasing cervical cancer screening, a few participants of this study were sceptical about the acceptability of self-sampling method by young girls and women, which is consistent to other similar studies conducted in low resource settings [24–26]. In contrast, a study conducted in Nepal which has analysed the feasibility of self-sampling among women reported more than 57% of women are willing to accept the self-sampling method if offered the service [15]. Similar to these findings, other studies conducted in different countries to explore feasibility and acceptability with end users showed that women are willing to accept the self-sampling approach to cervical cancer screening compared with clinician-based cervical cancer screening in the clinical settings due to social and cultural barriers, inconvenience and confidential barriers [27–31].

The low level of awareness on HPV, cervical cancer and self-sampling in the community is the result of low commitment and investment in community programs in Nepal In the context of Nepal reviews by Narsimhamurthy et al, Anderson et. Al and Darj et al suggest the need for increased investment in a community based program for cervical cancer screening, provision of information and improved service uptake in Nepal [8, 24, 26]. This study further builds

upon previous studies conducted and identifies the importance of investing in counselling, information, and awareness raising for self-sampling cervical cancer screening and its impact on women's social, psychological, and financial aspect of life. Similarly, this research suggests a national policy and national program for cervical cancer screening incorporating self-sampling is likely deliver improved cervical screening uptake and cancer mortality outcomes and reduce the public health burden.

Even though specialized services are covered financially under health insurance program [32], this study highlights the out-of-pocket expenditure as a barrier to accessing cervical cancer self-sampling service. Some participants in this study suggested cervical self-sampling as a costly approach to improving cervical screening coverage. However, a randomized clinical trial involving more than 36,000 women showed that self-sampling for HPV is more cost effective than the traditional PaP smear [33]. Systematic review and meta-analysis published in 2022 also showed self-sampling of HPV cost effective, considering lower costs for sampling kits and testing and higher sensitivity to detect cervical cancer. The study also emphasised the need for further studies on the cost-effectiveness of cervical cancer self-sampling in low- and middle-income settings [34].

The accuracy and quality of self-sampling, which could influence the results, have been a concern across the participants of this study. Although there is no such evidence from Nepal measuring the sensitivity and specificity, the 2018 meta-analysis study of 56 diagnostic test accuracy studies and 25 randomized trials, as well as a 2013 meta- analysis of eight European countries, Mexico and United State of America showed that self-sampling of HPV testing using polymerase chain reaction (PCR) - testing has similar sensitivity and specificity to clinical sampling [11, 35, 36].

The study by Shrestha et al. in Nepal in 2021 suggests that the women are willing to use the online delivery services for self-sampling [15], and the findings of this study complement the previous study where online service providers showcase ready for online delivery services. This finding indicates a preference for self-sampling service availability (such as pharmacies and online services) for cervical cancer screening. This study recommends establishing an effective and clear mechanism for the introduction of self-sampling as an option for cervical cancer screening in the Nepali national policy and guideline. For instance, in many high income countries, women undertake the cervical cancer self-sampling by mailing samples from their home [12]. Although participants of this study suggest scale up of self-sampling method for cervical screening in urban areas, it is equally important to reach rural girls and women. To achieve this, it is essential to disseminate the importance of self-sampling for cervical cancer screening and to explain the process of the self-care approach for cervical cancer screening. Additionally, this study also recommends national level studies to understand the acceptability, effectiveness, accuracy of self-sampling approach to guide national programs, guidelines, plans and activities for effective roll out of self-care approach in cervical cancer screening.

## Limitation and recommendations of the study

The study provides valuable insights into the feasibility and acceptability of HPV self-sampling for cervical cancer screening among health care providers and online service providers at federal level but does not explore the acceptability and feasibility of self-sampling approach among local government and online service providers, which is one of the limitations of this study. Thus, this study recommends further research to capture perspective of local health care providers, online service providers and policy makers to measure the convenience of self-sampling approach among women and girls, and assess the cost-effectiveness and accuracy of self-

sampling in Nepal which is crucial to guide national programs and policies for prioritization and effective implementation of self-sampling approach in cervical cancer screening.

## Conclusion

In conclusion, self- sampling for cervical cancer screening is a feasible and acceptable alternative service delivery for health care provider and policy makers in the federal level of Nepal. The barriers to prioritize, implement and scale up self-sampling of cervical cancer screening as identified by this study include limited knowledge and information about cervical cancer, screening related costs, not a clear understanding of the self-sampling process and the need for a centralized service delivery mechanism. Strategic and evidence-based decision making is essential to support the effective adoption and roll out of the self-sampling option of HPV among health care providers and policy makers in Nepal.

## Supporting information

**S1 Checklist. Inclusivity in global research.**
(DOCX)

**S2 Checklist. COREQ (COnsolidated criteria for REporting Qualitative research) checklist.**
(DOC)

**S1 Text. Interview guideline.**
(DOCX)

## Acknowledgments

The research team express gratitude to The Britain Nepal Medical Trust, UK and The Birat Nepal Medical Trust Nepal for their support to the project. Special thanks to Mr. Samip Pandey and Mr. Bikram Singh for their valuable assistance in the data collection. The team extends its appreciation to all the members of BNMT Nepal, as well as Dr. Maxine Caws and Ms. Anchal Thapa, for their support and guidance.

## Author Contributions

**Conceptualization:** Saki Thapa, Bikram Bucha.

**Data curation:** Saki Thapa, Swastika Shrestha.

**Formal analysis:** Saki Thapa, Swastika Shrestha.

**Investigation:** Saki Thapa, Swastika Shrestha.

**Methodology:** Saki Thapa, Swastika Shrestha.

**Project administration:** Saki Thapa, Bikram Bucha, Safal Kunwar, Raghu Dhital.

**Resources:** Saki Thapa.

**Software:** Saki Thapa, Amit Timilsina.

**Supervision:** Saki Thapa, Amit Timilsina, Raghu Dhital.

**Validation:** Saki Thapa, Amit Timilsina, Bikram Bucha.

**Visualization:** Saki Thapa, Amit Timilsina.

**Writing – original draft:** Saki Thapa.

**Writing – review & editing:** Saki Thapa, Amit Timilsina, Gillian Holdsworth.

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
