## [Decision Letter · Decision Letter 0]

27 May 2024

PGPH-D-24-00552

Are we ready for self-sampling for cervical cancer screening? Insights from service providers and policy makers in Nepal.

Dear Dr. Thapa,

Thank you for submitting your manuscript to PLOS Global Public Health. After careful consideration, we feel that it has merit but does not fully meet PLOS Global Public Health’s publication criteria as it currently stands. Therefore, we invite you to submit a revised version of the manuscript that addresses the points raised during the review process.

We look forward to receiving your revised manuscript.

Kind regards,

Lei Gao

Academic Editor

Journal Requirements:

Additional Editor Comments (if provided):

Reviewers' comments:

Reviewer's Responses to Questions

**Comments to the Author**

1. Does this manuscript meet PLOS Global Public Health’s publication criteria? Is the manuscript technically sound, and do the data support the conclusions? The manuscript must describe methodologically and ethically rigorous research with conclusions that are appropriately drawn based on the data presented.

Reviewer #1: Partly

Reviewer #2: Yes

Reviewer #3: Partly

2. Has the statistical analysis been performed appropriately and rigorously?

Reviewer #1: No

Reviewer #2: Yes

Reviewer #3: N/A

3. Have the authors made all data underlying the findings in their manuscript fully available (please refer to the Data Availability Statement at the start of the manuscript PDF file)?

Reviewer #1: Yes

Reviewer #2: Yes

Reviewer #3: Yes

4. Is the manuscript presented in an intelligible fashion and written in standard English?

Reviewer #1: Yes

Reviewer #2: Yes

Reviewer #3: No

5. Review Comments to the Author

Reviewer #1: I'd like to acknowledge the authors’ efforts in raising awareness about the importance of self-sampling to improve the acceptance of self-sampling for cervical cancer screening in Nepal. Unfortunately, some areas have not been thoroughly addressed, and there are still some confusing segments demanding review for improvement.

MAJOR CONCERNS:

1. Line 101-102: The authors mentioned settings including Kathmandu, Bhaktapur, and Lalitpur districts of Nepal. Unfortunately, only one was elaborated on for the reason of their choice.

2. Worryingly, the authors further mentioned, “Kathmandu Valley was purposively selected for the study sites due to the presence of tertiary level health facilities, digital/online service providers, and federal-level policymakers who are based in Kathmandu Valley.” This poses a significant concern as it introduces bias. This raises questions about the minds and opinions of girls in rural and low-resource settings. Unfortunately, the study is titled “…in Nepal,” which does not represent important opinions as required.

In other words, the statement suggests that the data cannot mirror the opinions of non-educated and/or rural women. What about awareness in this group?

3. Furthermore, in Lines 119-138, the statement, “Participants were provided with an information sheet detailing the study's goals and ethical considerations. Study participants were also given information about the self-sampling approach and how to use the self-sampling kit, and the kit was demonstrated during the interview,” raises concern due to the information released to the participants. How the authors would be able to filter the authenticity of their statements from personal thoughts is a big question.

4. Line 220-223; One of the points raised, “So the knowledge about screening is the main reason why its status is low. Most of the patients go for treatments only with a doctor’s advice or after their visit to a gynecologist. Based on my knowledge, the patients who visit solely for screening are the least,” raises more concern regarding personal opinions versus reality. A reasonable and inquisitive mind would ask how the opinions of uneducated countrywomen, as captured in this statement, actually represent the true scenario.

5. In addition to the above, the authors further caused confusion by mixing “self-sampling” with “self-screening” as reflected in their text;

a. Line 191-193 and some other areas, “The people in urban cities are mostly aware and educated. Not only in Kathmandu valley but also in other urban cities like- Narayanghat, Butwal, and Pokhara. If we promote it and inform about this new method then I believe people would perform self-screening easily”.

b. And Line 278-280: “The main thing for promotion is to spread public awareness of this cervical cancer test kit among people about how to do screening at home, who can test it, where to send the sample”. (Online service provider-1)”

6. Line 215-216: “a lack of privacy during pelvic examinations are highlighted as barriers by health care providers…” is not clear. One would like to know if the authors meant that sampling dor the screening is done in Nepal without proper enclosure.

7. Line 263-265; The statement “There may be cases where they find it difficult to take their own samples, they may be skeptical whether the sample they take will be good enough or not, or they might fear that what they are doing is probably not right”. (Representative DOHS-3) suggests that this kind of statement demands hearing some of these women directly......it raises concerns of accuracy, or honesty, personal emotion could affect the responses!!!

8. There is a lot of info glaringly missing from the rural part as evidenced even in the care providers' response, “Online purchase of self-sampling kit is possible in urban areas. If the client pays the transportation charge, we will provide it. We don't know the feasibility in the villages. But in the city, women do not shy away to buy the medical product online from our experience”.

9. Line 377; The authors mentioned, “This is the first study to explore the feasibility and its acceptability of HPV self- sampling in Nepal with service providers.” Unfortunately, Ref 24 suggesting the same group of authors had earlier conducted this kind of study before and proved the authors wrong. This is worrying!

10. ABSTRACT:

The statement reading “an unclear screening process” is unclear and therefore requires correction to maintain good flow.

11. All abbreviations using apostrophes such as “It's” should be written out in full in formal writing.

12. Several sentences taken from the literature are not properly cited, which is a grave scientific violation. A few of these are listed below:

a. Line 63: “The World Health Organization (WHO) recommends a well-organized system for effective screening, diagnosis, treatment, and follow-up of cervical cancer.”

b. Line 69: “The initial target was to screen at least 50% of women in the age group of 30–60 years, by 2015, revised in 2017 to 70%.” If this is a statement from the same authors as the subsequent sentence, both sentences should have been merged.

c. And many others.

1. Line 73: The use of the connector “However” within 2-3 sentences would benefit from being changed to words like “Unfortunately” to maintain good flow. “However, a systematic review and meta-analysis suggests low cervical cancer screening where only 17% of the cervical cancer screening occur in hospitals while just 16% of the screening takes place in the community using the Pap Smear test for cervical cancer screening in Nepal (7).”

2. Many grammatical errors and punctuation issues were noted.

GENERAL COMMENTS

I think the authors failed to clarify the difference between self-sampling as just a part of the screening process and self-sampling as the entire screening method. Their erratic reporting made the concept difficult to understand.

Of most significance is the introduction of bias by informing the participants about the project in detail. One does not need to be a trained psychologist to see that the opinions recorded in the study may not accurately reflect real scenarios. In fact, one of the statements mentioned a lack of knowledge among participants.

The authors did not provide a good rationale for their choice of study settings and how these settings represent the entire country. Although the title mentions Nepal, only 20 opinions were recorded, which is insufficient to represent the whole country. These opinions are partially disconnected from reality and fail to include rural scenarios.

Thank you.

Reviewer #2: Very well written paper. Easy to read and follow through.

Lines 55-57: Would you consider referencing GLOBOCAN 2022 and check if cancer of the cervix is not currently in the third place globally in women?

Line 57: Is Nepal a low- or middle-income country?

Lines 59 – 60: You need to briefly discuss the link between the cancer of the cervix and human papillomavirus infection. Also, what is the comparator globally i.e., what is the percentage of women harbouring HPV globally.

Lines 63 - 65: A brief mention of WHO’s position on HPV testing might be helpful. Further, its strategy is to screen 70% by 2030. Since you are writing for a global audience, it helps to bring in global context.

Line 67: What sort of age groups should be screened according to CCSP i.e., lower and upper age boundaries?

Line 71: “…Nepal lacks a national screening program for cervical cancer…..” but in line 67 you mention CCSP. Is this not contradictory?

Line 81: What is the reference, please?

Line 407: There is mention of out-of-pocket expenditure. What is the country’s funding program for medical expenses supposed to be like?

Line 412: Reference, please.

Is there any study you could reference that discusses reasons for not taking up the current screening methodologies by women in Nepal?

Did you pilot the instruments? It might be helpful to add that information here if you did?

How many female participants were in your study? Attitudes to self-sampling might differ by gender.

It would have been interesting to include the views of the laboratory technologists because they are on the receiving end, and they are the ones who analyse the samples. Hence, they are in a better position to comment about correctness and quality of self-collected samples.

Comment on whether women are charged for cervical cancer screening.

Is self-sampling already available?

• What are the current pathways and time frames from collecting the sample through to getting results?

• Is this process not a deterrent to cervical cancer screening?

• Who is getting it?

What are the limitations and strengths of this project?

Reviewer #3: I read with interest this study entitled “Are we ready for self-sampling for cervical cancer screening? Insights from service providers and policy makers in Nepal.” Overall, the study addresses a crucial topic in public health interventions, especially given the renewed momentum to eliminate cervical cancer and the global call to action to enhance key strategies for preventing human papillomavirus (HPV) infection and eliminating cervical cancer, including immunization, screening, and treatment. Specifically, the study examines the perceived acceptability and feasibility of HPV self-sampling among service providers and policymakers in Nepal through qualitative research methods. By exploring perceptions of alternative screening tools, namely HPV self-sampling, the study tackles a topic of significant relevance that could help overcome certain barriers to access cervical cancer screening. Moreover, this research aligns with global efforts to improve strategies for cervical cancer elimination. However, I noticed several areas in the manuscript that require significant attention and revision before publication. Therefore, I propose the authors consider the following comments:

General comments:

One of the main concerns I have pertains to the lack of clarity surrounding the study design. The methodological approach and theoretical frameworks (if any) guiding the study are not adequately described (e.g. exploratory or deductive, case study, grounded theory, phenomenology, etc.). Clarifying the research paradigms underpinning the study helps the reader understand how the researchers address the research topic, as well as the choice of methods, tools, and procedures, including the analytical approach and the frame through which findings are interpreted. This information is missing, although it is part of the COREQ guidelines that the authors claim to follow (lines 151-152). Moreover, presenting the perspectives on, or the definitions of “acceptability” and “feasibility” used in the study would allow readers to grasp how the results are interpreted, as these terms are widely used in global health and implementation research. Without clear definitions, results may overlap across thematic categories. For example, participants’ views on the difficulty of implementing self-testing in rural areas due to limited access to internet connection appear twice under different themes: feasibility/facilitators and perceptions. Additionally, both terms seem to be used interchangeably and, while the term “acceptability” does not appear as a theme (table 1), it is used several times within the text. Therefore, I strongly recommend the authors revise the definitions and make a critically use of the concepts, and also explore the relationship between them if appropriate (for example, see line 386: “Although service providers and policymakers find self-sampling to be a feasible process for increasing cervical cancer screening, a few participants of this study were sceptical about the acceptability of self-sampling method by young girls and women”).

Furthermore, the manuscript would benefit, overall, from more contextual information to enhance reader comprehension. Specifically, details on the study setting in the methods section, such as incidence rates or socio-demographic characteristics, would provide valuable background. Also, providing more details on the functioning of the HPV self-sampling method addressed in the study would help understand several participants' insights. In this vein, explaining why internet connection is important (line 188) according to participants would prevent the reader from making assumptions. This also helps in understanding the selection criteria of study sites (line 103: presence of digital/online service providers). Similarly, the aspect of online healthcare highlighted by some participants (line 320) might need more thorough description. In this vein, the background section could be slightly expanded to introduce some of the existing evidence related to the topic that could help outlining the research context and build the rationale of the study (as mentioned by the authors -line 87), feasibility studies for HPV self-sampling methods have been conducted).

Another concern pertains to the discussion of results, which does not align well with the results presented in the corresponding section. The discussion should accurately reflect the data collected and dialogue with existing evidence (previous works) to support the findings. Indeed, integrating the identified themes from the results section into the discussion would provide a coherent narrative and facilitate a deeper understanding of the implications of the findings.

This reviewer would recommend including some lines on limitations. For example, even if it is beyond the scope of the study, mentioning the lack of participation from women/end-users, which would have facilitated data triangulation, would be beneficial.

Finally, the text overall needs improvement in terms of readability and connection of ideas. A thorough revision is recommended to address typos and grammar mistakes, and to be more meticulous with the subtitles used.

Specific comments:

Introduction:

• In the first paragraph, although HPV-16/18 infection is mentioned, there is no explanation or reference to its role as the main cause of cervical cancer. Introducing this idea briefly would facilitate readability. Additionally, acronyms such as HPV should be spelled out in full the first time they are used.

• A reference to the WHO Global Strategy for Cervical Cancer Elimination would be beneficial, particularly when mentioning the target for women screened according to the national guidelines, since it is in line with the WHO global strategy.

• In line 84, ("the Government of Nepal highlights the challenge of a technology-based programme but emphasizes the importance of self-sampling in reaching larger populations in the reference manual for Cervical Cancer Screening in Nepal 2015”), more context could be included) so that the reader understands what specific challenges the Government of Nepal is referring to.

Methods

• The reference to purposive sampling sounds repetitive (it is mentioned twice - lines 98 and 106, though it is not needed in the study design paragraph).

• More details on the criteria used to select the study settings would be beneficial: why are tertiary level health facilities important within the context of this study?

• What are digital/online delivery services (lines 109, 128-129)?

• Lines 115-116: “Purposive sampling was adopted for data saturation”. Can the authors explain what they mean here?

• Ethical considerations usually go in a separate specific subsection within the methods section. Beyond the formatting aspect, I recommend not mixing this information with other content (see lines 112-122).

Results:

• (Lines 201-203): “However, some of the participants of this study suggest that self-sampling is not feasible and acceptable in the context of Nepal, preferring to support screening by health care providers, and anticipating potentially low participation from the service users”. Would it be possible to present the reasons expressed by these participants not to perceive it acceptable/feasible?

• Line 224: no need to spell out IDI again.

• There is a subsection named “Inclusive policy and national strategies” that does not appear in the list of themes identified through the thematic analysis, which seems confusing at this point. While qualitative analytical approaches can identify emerging themes outside the study's initial objectives, it would be beneficial to contextualize the reader when introducing unexpected findings.

Discussion

• Lines 381-383: “Convenience, confidentiality, and a low level of embarrassment have been identified as key characteristics of adopting self-sampling among the participants.” This sentence seems not to be aligned with the results presented in the previous section. Can the authors provide some clarifications?

• Lines 387-390: “…a few participants of this study were sceptical about the acceptability of self-sampling method by young girls and women, the myths and misconceptions regarding self-sampling approach among certain sections of society, particularly those of low literacy and poor socio-economic status”. There’s no mention of myths and misconceptions in the results, thus, it seems this is an assertion not supported by reported data.

I hope that above comments would help the authors to improve the manuscript. I am so looking forward to read the article once published

Kind Regards.

6. PLOS authors have the option to publish the peer review history of their article (what does this mean?). If published, this will include your full peer review and any attached files.

**Do you want your identity to be public for this peer review?** For information about this choice, including consent withdrawal, please see our Privacy Policy.

Reviewer #1: **Yes: **Kamoru Adedokun

Reviewer #2: No

Reviewer #3: No

---

## [Decision Letter · Decision Letter 1]

22 Sep 2024

PGPH-D-24-00552R1

Are we ready for self-sampling for cervical cancer screening? Insights from service providers and policy makers in Nepal.

Dear Dr. Thapa,

Thank you for submitting your manuscript to PLOS Global Public Health. After careful consideration, we feel that it has merit but does not fully meet PLOS Global Public Health’s publication criteria as it currently stands. Therefore, we invite you to submit a revised version of the manuscript that addresses the points raised during the review process.

We look forward to receiving your revised manuscript.

Kind regards,

Lei Gao

Academic Editor

Journal Requirements:

Additional Editor Comments (if provided):

Reviewers' comments:

Reviewer's Responses to Questions

**Comments to the Author**

1. If the authors have adequately addressed your comments raised in a previous round of review and you feel that this manuscript is now acceptable for publication, you may indicate that here to bypass the “Comments to the Author” section, enter your conflict of interest statement in the “Confidential to Editor” section, and submit your "Accept" recommendation.

Reviewer #1: All comments have been addressed

Reviewer #2: (No Response)

2. Does this manuscript meet PLOS Global Public Health’s publication criteria? Is the manuscript technically sound, and do the data support the conclusions? The manuscript must describe methodologically and ethically rigorous research with conclusions that are appropriately drawn based on the data presented.

Reviewer #1: Yes

Reviewer #2: Yes

3. Has the statistical analysis been performed appropriately and rigorously?

Reviewer #1: Yes

Reviewer #2: Yes

4. Have the authors made all data underlying the findings in their manuscript fully available (please refer to the Data Availability Statement at the start of the manuscript PDF file)?

Reviewer #1: Yes

Reviewer #2: Yes

5. Is the manuscript presented in an intelligible fashion and written in standard English?

Reviewer #1: Yes

Reviewer #2: Yes

6. Review Comments to the Author

Reviewer #1: None

Reviewer #2: Thank you for addressing comments. There are still a few more issues that need your attention, please.

General comments:

You need to be consistent about categories of participants. The way that I understand it is that you have: (i) healthcare providers (nurses and gynaecologists), (ii) service providers (online services) and (iii) policymakers. This is not always clear as you switch between two and three.

You have not included education/literacy as a barrier to self-sampling.

Further, you have not discussed understanding sampling processes as a barrier.

What is the strength of the study?

The recommendation is under limitations.

From your understanding, how would government subsidise this method for an average hospital attendee.

Also, make some mention of how gynaecological services are paid for in Nepal.

Please pay attention to grammar.

Specific comments:

Line 40: “Screening” should be “sampling”.

Line 43: “Women and girls” does this refer to study’s female participants i.e. study population or is it referring to users users?

Line 61: Papilloma virus is one word “papillomavirus”

Line 66: Reference?

Line 69: use “WHO” instead of the fully spelled out World Health Organization.

Line 71: Do not use capital letters for the first letter for Cervical Intraepithelial Neoplasia.

Line 74: Do not use capital letters for the first letter for Cervical Cancer Screening and Prevention.

Lines: 82-83: Not understandable.

Lines 82-88: Very long sentence. Break it down into shorter sentences.

Lines 94-97: Already alluded to in lines 64-66.

Line 99: Do not use capital letters for the first letter for Cervical Cancer Screening.

Line 127: “18 years and above” were those patients?

Lines 119-124: Should go under sampling i.e., the section below

Lines 139-141: Repetition. Further, they do not belong in this section.

Line 226: “Alternative to being sampled as a healthcare provider.”

Line 257: “Support sampling by”

Lines 281-282: Repetition. Delete.

Line 320: Consider saying “an unclear process between sample collection and delivery of results”

Lines 337-341: These lines could go to the section above.

Line 329: This could read, “Education about processes around sample collection”

Lines 364-367: Same quotation as in lines 302-306. This time it was by Gynaecologist 1 and not 3.

Lines 517-518: Is DHOS not local government. If not, then what is DHOS? This seems contradictory.

7. PLOS authors have the option to publish the peer review history of their article (what does this mean?). If published, this will include your full peer review and any attached files.

**Do you want your identity to be public for this peer review?** For information about this choice, including consent withdrawal, please see our Privacy Policy.

Reviewer #1: **Yes: **Kamoru A, Adedokun

Reviewer #2: No

---

## [Decision Letter · Decision Letter 2]

9 Dec 2024

Are we ready for self-sampling for cervical cancer screening? Insights from service providers and policy makers in Nepal.

PGPH-D-24-00552R2

Dear Ms Thapa,

We are pleased to inform you that your manuscript 'Are we ready for self-sampling for cervical cancer screening? Insights from service providers and policy makers in Nepal.' has been provisionally accepted for publication in PLOS Global Public Health.

Best regards,

Julia Robinson

Executive Editor

Reviewer Comments (if any, and for reference):

Reviewer's Responses to Questions

**Comments to the Author**

1. If the authors have adequately addressed your comments raised in a previous round of review and you feel that this manuscript is now acceptable for publication, you may indicate that here to bypass the “Comments to the Author” section, enter your conflict of interest statement in the “Confidential to Editor” section, and submit your "Accept" recommendation.

Reviewer #1: All comments have been addressed

Reviewer #2: All comments have been addressed

2. Does this manuscript meet PLOS Global Public Health’s publication criteria? Is the manuscript technically sound, and do the data support the conclusions? The manuscript must describe methodologically and ethically rigorous research with conclusions that are appropriately drawn based on the data presented.

Reviewer #1: Yes

Reviewer #2: Yes

3. Has the statistical analysis been performed appropriately and rigorously?

Reviewer #1: Yes

Reviewer #2: Yes

4. Have the authors made all data underlying the findings in their manuscript fully available (please refer to the Data Availability Statement at the start of the manuscript PDF file)?

Reviewer #1: Yes

Reviewer #2: Yes

5. Is the manuscript presented in an intelligible fashion and written in standard English?

Reviewer #1: Yes

Reviewer #2: Yes

6. Review Comments to the Author

Reviewer #1: None

Reviewer #2: Lines 169-170: What is the reference for? The thought is incomplete “Data collection

continued until reaching the point of saturation, where no new information (18).”

Lines 512 – 514: This section should go under recommendations below.

Limitation and Recommendations:

• Consider it as a limitation that only the affluent were included in the study – who talks for the illiterate women in the rural areas?

• Surely your study has strengths.

• Also, online service providers appear twice: under both local and federal government. Was this what you aimed at?

7. PLOS authors have the option to publish the peer review history of their article (what does this mean?). If published, this will include your full peer review and any attached files.

**Do you want your identity to be public for this peer review?** For information about this choice, including consent withdrawal, please see our Privacy Policy.

Reviewer #1: **Yes: **Kamoru Adedokun

Reviewer #2: No
